# Temporal Shifts in MicroRNAs Signify the Inflammatory State of Primary Murine Microglial Cells

**DOI:** 10.3390/ijms26125677

**Published:** 2025-06-13

**Authors:** Keren Zohar, Elyad Lezmi, Fanny Reichert, Tsiona Eliyahu, Shlomo Rotshenker, Marta Weinstock, Michal Linial

**Affiliations:** 1Department of Biological Chemistry, Institute of Life Science, Faculty of Science, The Hebrew University of Jerusalem, Jerusalem 91904, Israel; keren.zohar@mail.huji.ac.il (K.Z.);; 2Department of Genetics, Institute of Life Science, Faculty of Science, The Hebrew University of Jerusalem, Jerusalem 91904, Israel; 3Department of Medical Neurobiology, Institute for Medical Research Israel-Canada (IMRIC), Faculty of Medicine, The Hebrew University of Jerusalem, Jerusalem 91121, Israel; funarei@gmail.com (F.R.); shlomor@ekmd.huji.ac.il (S.R.); 4Institute of Drug Research, School of Pharmacy, The Hebrew University of Jerusalem, Jerusalem 91120, Israel; martar@ekmd.huji.ac.il

**Keywords:** innate immune system, IL-1, RNA-seq, purinergic receptor, inflammation, cytokines, miRBase, TarBase, CLIP-Seq

## Abstract

The primary function of microglia is to maintain brain homeostasis. In neurodegenerative diseases like Alzheimer’s, microglia contribute to neurotoxicity and inflammation. In this study, we exposed neonatal murine primary microglial cultures to stimuli mimicking pathogens, injury, or toxins. Treatment with benzoyl ATP (bzATP) and lipopolysaccharide (LPS) triggered a coordinated increase in interleukin and chemokine expression. We analyzed statistically significant differentially expressed microRNAs (DEMs) at 3 and 8 h post-activation, identifying 33 and 57 DEMs, respectively. Notably, miR-155, miR-132, miR-3473e, miR-222, and miR-146b showed strong temporal regulation, while miR-3963 was sharply downregulated by bzATP. These DEMs regulate inflammatory pathways, including TNFα and NFκB signaling. We also examined the effect of ladostigil, a neuroprotective agent known to reduce oxidative stress and inflammation. At 8 h post-activation, ladostigil induced upregulation of anti-inflammatory miRNAs, such as miR-27a, miR-27b, and miR-23b. Our findings suggest that miRNA profiles reflect microglial responses to inflammatory cues and that ladostigil modulates these responses. This model of controlled microglial activation offers a powerful tool with which to study inflammation in the aging brain and the progression of neurodegenerative diseases.

## 1. Introduction

Neuroinflammation is a key factor in neurodegenerative diseases (NDDs). Diseases such as Alzheimer’s (AD), Parkinson’s (PD), multiple sclerosis (MS), and amyotrophic lateral sclerosis (ALS) are characterized by chronic inflammation, mitochondrial dysfunction, and neuronal degeneration [1,2]. Microglia, the resident immune cells of the CNS, play a central role in responding to injury and pathological stimuli [3]. In AD, the prolonged activation of microglia leads to β-amyloid clearance and synaptic loss, while the release of inflammatory mediators from microglia causes neuronal damage [4,5,6]. The transition of microglia from a homeostatic to an activated state in response to stressors like brain injury or pathogens [7,8] is coupled with a morphological shift and transformation from ramified to amoeboid forms [9]. Activated microglia are categorized into pro- and anti-inflammatory states, with chronic activation leading to excessive secretion of pro-inflammatory cytokines, contributing to NDD pathology [10,11].

Taken together, the parameters of the inflammatory response are useful markers for disease progression [12,13]. During a chronic state, microglia secrete excessive pro-inflammatory cytokines (such as TNF-α, IL-1β, and IL-6) and reactive oxygen species (ROS). The molecular characterization of murine microglia has identified microglial-specific transcripts (~100) that correspond to their cell identity. Among them, *Tmem119*, *P2ry12*, *Siglech*, and *Cx3cr1* are highly expressed in resting microglia and are considered part of the homeostatic gene set. Microglial activation induces molecular markers (e.g., *Iba1, Cd68, Apoe*, *Lpl*, and *Cst7*) together with inflammatory genes such as *Ccl2, Il1b,* and *Nos2* [11]. The transcriptomic analysis of single-cell sequencing has expanded the list of marker genes and improved our understanding of the molecular signature underlying microglial functional states [14].

Studies on primary microglial cultures stimulated with ATP (bzATP) and lipopolysaccharides (LPS) have revealed coordinated inflammatory gene expression, including activation of the TNFα and NF-κB pathways [15]. Ladostigil, a neuroprotective compound, has been shown to reduce oxidative stress, inflammation, and cognitive decline in aging rodent brains [16]. Ladostigil restored the lowered mitochondrial potential induced in cells by H_2_O_2_ and decreased markers of oxidative stress. In microglial cultures, ladostigil inhibited the nuclear translocation of EGR1 and NF-κB complex, the phosphorylation of ERK1/2 and p38 proteins, and the release of pro-inflammatory cytokines [17]. While cell lines like BV2 and S9 serve as models, primary microglia better reflect in vivo responses [18].

In this study, we focused on the profile of microRNAs (miRNAs) in microglial cells. miRNAs are emerging as biomarkers for neurodegenerative diseases (NDDs) due to their role in regulating inflammation and oxidative stress, with potential applications in disease monitoring [19,20]. However, it is not yet clear how the modulated microglial activation selectively preserves their neuroprotective functions [21]. This study examines the changes in the miRNA transcriptomic profile upon stimulation and in the presence of ladostigil. We monitor the dynamics of the activation process by focusing on the differentially expressed miRNAs (DEMs) and discuss these miRNAs as potential indicators of microglial inflammatory states.

## 2. Results

### 2.1. Release of Pro-Inflammatory Cytokines in the Response of Microglia to Activation

Microglial function was assessed by quantifying TNFα and IL-6 secretion after stimulation, in the presence of LPS (0.75 µg/mL) and BzATP (400 µM) (Figure 1A; see Section 4). The baseline of secreted cytokines in untreated cells was below detection levels. TNFα and IL6 were detected 8 and 24 h after activation. At 8 h post-activation, the absolute levels of TNF-α and IL-6 were approximately 430 and 6 pg/μg, respectively. We normalized the amount of protein released (marked as 100%) for the TNFα and IL6 following bzATP/LPS. Testing the responsiveness of the culture, we exposed cells to budesonide, a synthetic glucocorticoid that produces anti-inflammatory and immunosuppressive effects by binding to cytoplasmic glucocorticoid receptors (GRs). The anti-inflammatory effect was detected already at 1 × 10^−13^ M for TNFα and IL6. At all budesonide tested concentrations, the reductions in IL6 were greater than those for TNFα (Figure 1A), and the effect was even stronger at 24 h (Figure 1B).

Inspecting the RNA transcript levels for Il6, Tnf, and IL1b showed very high induction already at 3 h post-activation with bzATP/LPS. Only Il6 exhibited a continuous increase in expression (Figure 1C). Tnfa displayed a transient trend, with a maximum level at 3 h (Figure 1C). We also analyzed the increase in the transcript of Il1b, a pro-inflammatory cytokine that plays a role in initiating and amplifying inflammation. The expression level peaked at 3 h post-activation and remained high also at 8 h (Figure 1C). We concluded that an abrupt and coordinated response to bzATP/LPS activation occurs within a time frame of a few hours. While monitoring mRNA levels is valid within 3–8 h post-activation, measuring the secreted proteins requires a longer period for completing ribosomal translation, folding, post-translational modifications, cell trafficking, and secretion. Although the activation kinetics for the different cytokines vary (Figure 1C), the time window of 3 to 8 h following bzATP/LPS activation captures the induction of the transcriptional cellular response. The increase in the level of secreted IL-1b was quantified in the microglial culture as detailed before [17]. These findings indicate that the pro-inflammatory response in the primary microglia culture is robust and coordinated.

### 2.2. Temporal Expression miRNA Profiles Following bzATP/LPS Activation

We sought to observe the temporal behavior of activated cells via the change in the miRNA profiles. Altogether, we identified 372 miRNAs that were stably identified (above a predetermined expression threshold, see Section 4) at 3 and 8 h after exposure to BzATP. Among all 372 miRNAs, only three were statistically significant (i.e., DEMs): miR-146b-3p and miR-146b-5p were upregulated to a level of 1.66 and 1.41 relative to untreated cells (N.T.), respectively, while miR-3963 was downregulated. Notably, the degree of the downregulation of miR-3963 was 5.8-fold in the presence of bzATP/LPS after 8 h, but did not reach the maximal level shown by bzATP alone (10.9-fold). From these results, we conclude that the addition of bzATP alone was unable to activate the inflammatory response of microglial cells, but potentially primes miR-146b and suppresses miR-3963.

The rest of the analyses were performed using miRNA-seq results from cells that were subjected to the activation protocol combined with bzATP and LPS (bzATP/LPS). We tested the cells’ miRNA transcriptome at 3 h and 8 h after exposure. The experimental groups were separated using unsupervised clustering based on miRNA expression data (Appendix A). Altogether, we identified 372 miRNAs that represent 345 uniquely labeled items from the miRNA-seq analysis. We measured the fold change relative to untreated cells and set a relaxed threshold on the fold change (log_2_FC > |0.33|). Each identified miRNA was assigned by its expression trend relative to that in the untreated cells and also to the expression level monitored at a previous time point.

Figure 2A shows the partitions of miRNAs into modules according to their expression patterns. We divided all miRNAs into nine expression patterns by their paired expression at the time points (expression trends are defined as in Appendix A). Each cluster is indicated by the number of miRNAs that match the expression pattern. As expected, most miRNAs (59%) were marked as unchanged even after 8 h (labeled ‘same–same’). Another 14% were characterized by a delayed response (i.e., labeled as ‘same-up’ and ‘same-down’; Figure 2A, black color). We found that 18% of the miRNAs were already upregulated (3 h post-activation), with more of them displaying a transient expression wave (Figure 2A, orange color). Only six miRNAs exhibited a consistent and robust increase in expression (Figure 2B). These miRNAs are expected to carry the signature for maintaining the activated state of the microglia culture. The identified miRNAs include miR-146b, miR-155-5p, miR-29b-2, miR-5121, and miR-6240. For 11% of the miRNAs, expression was suppressed compared to that in naïve cells (Figure 2A, blue color), and the expression of miR-301a and miR-760 was monotonically downregulated (Figure 2C). We conclude that many miRNAs are subjected to temporal regulation during the initial hours by bzATP/LPS. Expression modules for all miRNAs are listed in Appendix A.

### 2.3. Activation of Microglia by bzATP/LPS Alters the miRNAs’ Expression Profiles, Including Those of Abundant miRNAs

The cellular effects of miRNAs are tightly coupled to their quantities within cells [22]. Appendix A lists 20 miRNAs that are highly expressed (>100 CPM) and were significantly changed 8 h post-activation by bzATP/LPS. The list shows miRNAs by their mature variants, ranked by the fold change relative to untreated (N.T.) culture. Most of the miRNAs were upregulated, with the maximal fold change seen for miR-155-5p. The expression of the most abundant miRNA, miR-21a-5p (accounts for 16.2% of all identified miRNAs), was strongly upregulated (1.59-fold). It is anticipated that even a moderate increase in the expression of an abundant miRNA (albeit by ~60%) may indirectly affect the stability of other miRNAs, thus impacting global cell regulation [22]. Results from Appendix A emphasize the specificity towards the altered profile of miRNAs following activation. Specifically, although there are 20 miRNAs that belong to the let-7 family (accounting for >30% of all miRNAs in the microglial cells), they remain unchanged, with an exception being the downregulation of let-7b-5p. We concluded that the in vitro activation of the primary neonatal microglial culture is specific, resulting in substantial changes in miRNA profiles involving 15% of the abundant miRNAs (≥100 CPM). Also, among these differentially expressed miRNAs (DEMs) are highly abundant miRNAs such as miR-21a, miR-146b, and miR-7a.

Applying the same analysis for the miRNAs 3 h post-activation identified only 10 upregulated miRNAs, and none were downregulated. Among these DEMs we identified miR-21a and miR-146b, as well as abundant miRNAs that are only significant at the early phase of activation (e.g., miR-125a, miR-125b; Appendix A). We conclude that the establishment of fully activated microglia is reflected by the temporal specificity in miRNA profiles.

### 2.4. Dynamics of Differentially Expressed miRNAs (DEMs) by bzATP/LPS Activation

Activation of the primary microglial culture by bzATP/LPS changed the expression profiles of miRNAs. Of the 372 mapped miRNAs that were expressed in substantial amounts, 38.8% are expressed with ≥100 CPM (counts per million). Figure 3A shows that most miRNAs have a low expression, with 50% of the miRNAs below 50 CPM and only 14 with >20,000 CPM. This set accounts for more than 73% of all cellular miRNAs (Appendix A). Figure 3B marks the number of DEMs relative to their expression level. In microglia, similar to other cells, the miRNA expression level (y-axis) ranges over five orders of magnitude. Multiple variants of the same miRNAs dominate DEMs (e.g., miR-146, miR-155, miR-132, and others).

In all cells, the post-transcriptional regulation mediated by miRNAs upon translational arrest is expected to be fast and transcriptionally independent [23]. We therefore tested the dynamics of miRNAs at two time points following stimulation by bzATP/LPS (Figure 3). The normalized expression levels of all identified miRNAs are listed in Appendix A. There were 33 and 57 DEMs among the 372 miRNAs that met the statistically significant threshold (FDR ≤ 0.05) for the short-term (3 h, Figure 3C) and long-term (8 h, Figure 3D) activation protocols, respectively.

### 2.5. A Small Set of Temporally Responsive miRNAs Dictates the Establishment of the Fully Activated Microglia

Inspection of the results from the miRNAs that are consistently upregulated according to the set of time-dependent unique DEMs allowed us to infer the contribution of miRNAs to establishing the activated microglial state but also highlighted examples of transient expression. Figure 4 shows the DEMs along the activation timeline. Figure 4A presents a Venn diagram with most DEMs identified for the 3 h and 8 h post-activation paradigms (Appendix A). Only nine miRNAs are unique to the early time point. Furthermore, half of the DEMs at 8 h are also unique (27 DEMs), with similar amounts of up-and downregulation. Figure 4B (left) illustrates examples of DEMs with a maximal fold change at the early time point. Figure 4B (right) illustrates an opposite trend with fold change at a later time point that continues to increase. Examples are miR-132-5p and miR-155-3p, which show maximal expression at 3 h post-activation and also remain substantially high at 8 h post-activation, while miR-155-5p and miR-146b are maximal at 8 h post-activation. These miRNAs exhibit strong temporal sensitivity (≤30% difference between the two time points). We concluded that these miRNAs are of special interest as their levels may dictate a time-dependent regulation of the microglial culture that was shifted to new inflammatory states.

Figure 4C displays an MA plot of time-dependent fold change versus the mean expression post-activation (Appendix A). We identified 15 miRNAs (4%) that are temporal DEMs (T-DEMs), with 5 that are maximally changed at the 3 h time point (marked red) and 10 that were further altered at a later time point (8 h, colored blue). The direct time-dependent comparison highlighted several miRNAs that are more sensitive to the dynamics of the activation process. This includes miR-455-5p, mir-365-2-5p, and miR-139-3p. An opposite trend (i.e., maximal expression occurs at 8 h post-activation) included the following T-DEMs: miR-146b-3p, miR-664-3p, miR-671-3p, miR-155-5p, miR-7a-5p, miR-146a-5p, and miR-365-3p. We observed that miR-365 showed a temporal dynamic, while different variants of miR-365 exhibited an opposite temporal trend (miR-365-2-5p and miR-365-p).

### 2.6. Temporal Expression of a Set of miRNAs Is Coupled with Differentially Expressed Inflammatory Genes

The exposure of the cells to bzATP/LPS led to a group of miRNAs changing their expression (Figure 4C). We tested whether these T-DEMs affected the establishment of the microglial inflammatory state. We mapped miRNAs to their appropriate targets and limited our analysis to miRNA-mRNA pairs that were experimentally validated. The list of T-DEMs (Appendix A) includes miRNAs that are classified as early and late responders, based on the time point of their maximal expression (3 h or 8 h post-activation). We merged T-DEMs with RNA-seq data that were collected from microglia under identical activation conditions [15] and used the miRNet 2.0 platform due to its multiple-modality capability.

The list of mRNA-seq results included 7970 genes (FDR ≤ 0.05, filtered for coding genes), among which 149 were identified as direct targets in microglia (Appendix A). The rest of the genes were either not expressed in microglia or failed our thresholds. We focused on the 25 genes with a substantial temporal expression change (T-DEGs with log2(FC) < |1|). All listed miRNA–target pairs shown were validated experimentally and were identified microglia as T-DEGs (Table 1).

Table 1 lists 21 of these miRNA–targets (filtered by an expression threshold of ≥10 CPM) along with their statistical properties. Several observations emerged regarding the potential impact of miRNAs on the direct targets: (i) Most listed miRNAs are abundant (>200 CPM, bold). Extremely abundant miRNAs include miR-7a-5p and miR-146a-5p, with expressed levels of 16,738 and 11,147 CPM, respectively. (ii) miR-155 is associated with many of the targets (11 of 21). (iii) Some of the target genes are very highly expressed in microglia. Examples include Tnf (CPM 5250.7), Nlrp3 (CPM 1000), and Nfkb1 (CPM 533.3). (iv) Among the listed targets, five were strongly upregulated compared to naïve cells (DEMs with log2(FC) < |1|), with Nos2 and Tnfa showing fold changes of 29.13 and 18.92, respectively.

Figure 5A shows the results of miRNet 2.0, centered around a small set of T-DEMs. It is evident that miR-155-5p is a major hub, with many paired mRNA genes, and some genes are regulated by more than one miRNA. Revisiting the results from miR-155-5p shows that it is strongly upregulated (22-fold) just 3 h post-activation and remains high at 8 h.

Figure 5B shows that Tnf is a strong hub with Il6ra, Nlrp3, Nos 2, and more. These transcripts are strongly expressed following microglial activation. The assembly of the NLRP3 inflammasome leads to the release of pro-inflammatory cytokines, such as IL-1β. In microglial cells, miR-7a-5p is highly expressed and known to play a role in modulating neuroinflammation by directly targeting Nlrp3, resulting in reduced pro-inflammatory cytokine production. Over half of all targeted genes to T-DEMs are annotated as ‘regulation of cytokine production’ (GO:0001817; enrichment *p*-value 3.4 × 10^−6^).

While Figure 5A presents overall interactions of miRNAs with validated target genes from any tissue, Figure 5B shows the protein–protein interaction (PPI) network of target T-DEGs, as identified by RNA-seq from bzATP/LPS activated microglia [15]. Many of the listed genes belong to cytokine release and immune-related categories. We further tested the miRNA–target pairs (Table 1) by examining the pathways represented in Reactome. Figure 5C presented an enrichment analysis for the set of pathways that dominate the input genes. We show that cytokine signaling, interleukins, and TCR pathways are very significant. Other pathways indicate that a large fraction of the input gene list was implicated in the NLR3 inflammasome.

### 2.7. Ladostigil Induced miRNAs That May Serve as Mediators in Suppressing Inflammation

The analysis of miRNA profiles in the activation protocol indicated that the system is suitable for pharmacological manipulation and analysis [24]. Accordingly, we examined the effect of ladostigil on miRNA profiles. Ladostigil, an aminoindan derivative [25], has been shown to reduce the production and secretion of pro-inflammatory cytokines [17]. RNA-seq analysis revealed that Egr1, Egr2 (Early growth response protein 1 and 2), and several metalloproteinases (MMPs) were upregulated following the microglia activation protocol, while the incubation of ladostigil significantly reversed this trend [17,24]. We determined whether miRNAs could be the mediators controlling the overall reduction in the inflammatory state of microglia. miRNAs may exert their function through several modes of action, including cellular relocation [26], loading into AGO proteins for stabilization, and indirect competition with other RNAs [27]. In a short time window, the miRNA profile was compared between N.T. (2 h of incubation with ladostigil prior to activation) and 3 h following full activation. We found that none of the 372 identified miRNAs were significantly altered (Figure 6A). We concluded that miRNA expression changes were not involved in the immediate response to ladostigil. However, 8 h with ladostigil moderately upregulated four miRNAs: miR-23b-5p (1.48-fold), miR-27a-5p, miR-27b-5p (1.27–1.28-fold; Figure 6B, Appendix A), and miR-365-2-5p (1.93-fold). While the role of miR-365-2-5p is not known, the other miRNAs have previously been implicated in inflammation suppression (Table 2). Despite the relatively moderate fold changes in miR-27a and miR-27b, they are highly abundant and thus may be involved in the suppression of multiple targets (their average amounts in microglia are 532.2 and 167.5 CPM, respectively).

Table 2 summarizes the current knowledge on the link between the overexpression of miRNAs and their potential targets that drive inflammation reduction. We conclude that ladostigil induces a restricted set of miRNAs that enhance the ability of the cells to cope with oxidative stress and the induction of pro-inflammatory signature of the treated cells. The central role of miRNAs in attenuating NF-kB signaling (Table 2) is in accord with the significance of this pathway in the modulation by ladostigil in other cellular systems (e.g., [34]).

## 3. Discussion

Our study focused on changes in miRNAs in primary neonatal purified cultures of microglia that exhibit a strong response to external signals of bzATP/LPS. These stimuli mimic the microglial microenvironment upon exposure to pathogens causing substantial cell death. The contribution of miRNAs to the inflammatory environment has been extensively studied in neurodegenerative diseases (NDDs). Dysregulation of miRNAs in microglia has been reported in an ALS mouse model (with mutated *Sod1*), along with their response to inflammatory signals, including microglial NLRP3 inflammasome activation [37]. miR-365 and miR-125b were among the miRNAs significantly upregulated in ALS microglia [38]. That study showed that miR-365 interferes with the IL-6 pathway, while miR-125b affects the STAT3 signaling pathway. These interactions led to the increased production of TNFα, contributing to neurodegeneration in ALS. Similar to our findings, in the ALS model, miR-22, miR-155, miR-125b, and miR-146b were upregulated [38].

We found that radical change in the expression of miR-155 signifies microglial activation by bzATP/LPS and is directly associated with cytokine release, the hallmark of the microglial response to pathogens. Our results are consistent with the centrality of miR-155 in the inflammatory phenotype of M1 macrophages [39,40]. Using real-time PCR, it was confirmed that miR-155 contributes to the suppression of mRNA targets that lead to the induction of iNOS, IL-1β, TNFα, IL-6, and IL-12. Among the direct targets of miR-155 are Inpp5d, Ptprj, and other transcripts that were also identified in our study as T-DEGs (Table 1). The direct targets of miR-155 include Socs1 (suppressor of cytokine signaling 1) and Inpp5d (inositol polyphosphate-5-phosphatase D), leading to enhanced NF-κB signaling and inflammation. In addition, we showed that Tnf acts as a hub in the inflammatory network (Figure 5B). Tnf expression at 8 h is 3.4-fold higher than at 3 h. It is paired with T-DEM miR-132-5p, which is maximally expressed 3 h after exposing the cells to bzATP/LPS (Table 1). miR-132 is highly conserved between humans and mice and has been shown to be a regulator of neural signaling, where its dysregulation contributes to axonal damage. The miR-132-5p negatively regulates the release of TNF, possibly by targeting upstream regulators like NF-κB [41]. Although speculative, it is likely that the presence of miR-132-5p at early time points, and its reduced amounts at later stages, regulates the delayed accumulation of Tnf transcripts in activated microglial cultures. We also show that miR-146a-5p is paired with Nos2 (nitric oxide synthase 2, Table 1). The overexpression of miR-146a-5p in the BV2 microglial cell line reduced the expression of iNOS, encoded by the *Nos2* gene [42]. A similar effect was observed in miR-146a knockout mice, which resulted in the overproduction of pro-inflammatory cytokines (e.g., IL-1β, TNFα, and IL-6) [43]. These results confirm that miR-146a-5p acts to attenuate the microglial inflammatory state.

The contribution of chronic neuroinflammation to major NDDs, including AD, PD, ALS, and multiple sclerosis (MS), is reflected by the miRNA signature from microglia [44]. We suggest that the temporal analysis of cellular miRNAs can be utilized as a sensitive indicator for NDD progression and also to develop new therapeutic strategies for modulating the neuroinflammatory state. Chronic treatment with ladostigil in aging rats attenuated certain aspects of neuroinflammation, including the upregulation of the Adora2 ATP receptor [9]. In microglial cultures, activation by bzATP/LPS led to the upregulation of Egr1, Egr2, and PDGF-β, which are linked to P2X7R signaling and neuroinflammation [17]. The upregulation of transcripts such as TNFαIP3 by ladostigil highlighted its role in suppressing oxidative stress and MAPK pathways.

We suggest that the observed miRNA signature supports the importance of ladostigil in counteracting neuroinflammatory processes, possibly through the suppression of oxidative damage [24]. Such regulation by ladostigil was validated in the aging rat brain and in the prevention of memory decline. We show that the specific miRNAs that were induced by ladostigil were shown in model organisms and cellular systems to suppress inflammation, most likely by activating NF-kB signaling (Table 2). Ladostigil upregulated miRNAs that act to reduce inflammation; specifically, miR-27b counteracted the effects of TNFα and was linked to restoring mitochondrial function, reducing apoptosis, and the Akt/Foxo1 pathway [33]. Others have shown that changes in miR-23b are associated with PD [44], and have been implicated in the oxidative stress response [45] as well as suppressing α-synuclein expression. In mice with induced sepsis, miR-23b-5p was reduced, coupled with an elevation in ADAM10 and other MMPs. The expression of metalloproteinases increased in the activated microglia and was suppressed by ladostigil, alongside seeing a decrease in the levels of inflammatory cytokines (TNF-α, IL-1β, and IL-6) [17].

An apparent limitation of our cellular system is the lack of any crosstalk between the microglial culture and other cell types. Cell communication is expected to occur between microglia, neurons, and other cells in the central nervous system (CNS), such as astrocytes, endothelial cells, and stem cells. Exosomes and additional types of extracellular vesicles (EVs) serve as an added layer of cell communication [46]. Many of the miRNAs reported in this study were identified as circular miRNAs (e.g., in serum, plasma, and CSF) across a wide range of CNS disorders [47]. For example, in mouse models, astrocyte-derived exosomes were shown to regulate microglial responses following traumatic brain injury (TBI). Isolated exosomes carry miR-873a-5p, which inhibited ERK and NF-κB signaling in microglia, thus reducing neuroinflammation and accelerating the repair process [48]. Increased levels of EV-associated miRNAs, such as miR-21, miR-146, miR-7a, and miR-7b, were found in injured mouse brains [49]. It has been suggested that the source of miR-21 is neurons near the lesion site, and, consequently, that EVs alter the response of microglia. In our system, a similar set of miRNAs was upregulated by bzATP/LPS without any trigger from neurons or other cell types. We suggest that even a moderate increase in the expression levels of miR-21, miR-146a, miR-146b, and miR-7a can cause a shift in cellular miRNA regulation due to their extreme abundance. Changes in the relative abundance of miRNAs may activate an indirect effect on the availability of many other less-abundant miRNAs [22].

Exosome-based communication between resting and activated microglia was validated in a mouse model of retinal angiogenesis. It was confirmed that miR-155-5p within the exosomes caused the activation of the NF-κB pathway [50]. In another system, miR-181a-3p from mesenchymal stem cells (MSCs) affected oxidative stress in PD. Using SH-SY5Y neuroblast-like cells as a model for drug-induced PD, it was shown that miR-181a-3p was transferred via EVs from MSCs to the SH-SY5Y cells, where it affected the p38 MAPK pathway by inhibiting EGR1. We argue that the attenuation of oxidative stress underpins the mechanism of action of ladostigil [24], which can be manifested directly or through EV-mediated miRNA cellular communication. The regulatory impact of miRNAs within EVs on intercellular communication is strongly dependent on amounts and targets’ stoichiometry [51]. Currently, circular miRNAs and the molecular content of exosomes are considered attractive biomarkers. Observing a robust temporal expression of miRNAs following external stimuli suggests that profiling secreted miRNAs can be used as non-invasive indicators of cellular inflammatory states.

Our findings shed light on the miRNA–mRNA regulatory networks that govern the inflammatory state of primary neonatal microglia, highlighting potential therapeutic targets. The dysregulation of miRNAs, particularly the most abundant ones, suggests their potential utility as early biomarkers of neuronal damage in neurodegenerative diseases (NDDs) and other brain pathologies.

## 4. Materials and Methods

### 4.1. Compounds and Reagents

The cell culture reagents, including Dulbecco’s Modified Eagle Medium (DMEM), DMEM/F12, gentamycin sulfate, and L-glutamine, were obtained from Biological Industries (Beit-Haemek, Israel). The activation protocol included stable ATP 2′-3′-O-(4-benzoyl benzoyl), adenosine 5′-triphosphate (BzATP), and lipopolysaccharide (LPS) from *Escherichia coli* 055:B5, purified by trichloracetic acid extraction (Sigma-Aldrich, Jerusalem, Israel). Ladostigil was a gift from Spero Biopharma (Jerusalem, Israel).

### 4.2. Preparation of Microglial Cultures

Primary microglia were prepared according to a previously described protocol [17]. The cells were isolated from the brains of neonatal male Balb/C mice (Harlan Sprague Dawley Inc., Jerusalem, Israel). Briefly, cells were isolated and plated for 1 week in poly-L-lysine-coated flasks. Following a dissociation protocol, non-adherent and loosely adhered cells were re-plated for 1 h on uncoated bacteriological plates, which allowed for the removal of cells exhibiting slower adherence kinetics. Microglial cells were propagated and supplemented with a conditioned medium containing mouse-CSF (colony-stimulating factor) and maintained in heat-inactivated fetal calf serum (FCS). The purity of the culture was confirmed by morphological criteria and a set of markers as previously described [15]. The cultures are estimated to be >95% microglia-pure. Under such conditions, the microglial culture remains responsive for four weeks.

### 4.3. Measurement of Cytokines

For cytokine release experiments, the microglial culture was washed and the medium was replaced with purified BSA for 24 h. Data presented are for microglial cells stimulated by a combination of bzATP (400 µM) and LPS (0.75 µg/mL). We showed that the concentration of LPS (0.75 µg/mL) given together with BzATP did not affect cell viability after 3 and 24 h using the MTT assay as previously described [17]. Cell lysates were tested and normalized by protein levels using a BCA Protein Assay (Pierce, Meridian, Rockford, IL, USA), and a cytokine ELISA assay was performed according to the manufacturer’s protocols. Cells were grown to 75% confluence in 6-well plates. Measurements of cytokine secretion were made 24 h after activation in the presence of BSA (0.4 µM) using Max deluxe (Biolegend, San Diego, CA, USA) commercial ELISA kits. Assays for cytokine release were conducted with 5 × 105 cells per well and cytokine assays were calibrated by using internal standard curves.

### 4.4. MicroRNA-Seq

Microglial cultures were harvested using a cell scraper. Total RNA was purified from ~106 cells using QIAzol Lysis Reagent RNeasy plus Universal Mini Kit (QIAGEN, GmbH, Hilden, Germany). To ensure homogenization, a QIAshredder (QIAGEN, GmbH, Hilden, Germany) mini-spin column was used. Samples were transferred to an RNeasy Mini spin column and centrifuged for 15 s at 8000× *g* at room temperature. The mixture was processed according to the manufacturer’s standard protocol. Samples with an RNA integrity number (RIN) > 9, as measured by an Agilent 2100 Bioanalyzer, were considered for further analysis. RNA libraries (in triplicates) were prepared according to a NEBNext Small RNA Library Prep Set for Illumina (Multiplex Compatible) Library Preparation Manual. Adaptors were then ligated to the 5′ and 3′ ends of the RNA, and cDNA was prepared from the ligated RNA and amplified to prepare the sequencing library. The amplified sequences were purified on E-Gel EX 4% Agarose gels (ThermoFisher, Waltham, MA, USA, # G401004), and sequences representing RNA smaller than 200 nt were extracted from the gel. The library was sequenced using an Illumina (San Diego, CA, USA) NextSeq 500 Analyzer.

### 4.5. Bioinformatic Analysis and Statistics

All next-generation sequencing data underwent quality control using FastQC, version 0.11.9. Adapter trimming and quality filtering were performed using Trimmomatic, version 0.39, with the AllTrueSeqPE adapter list and a minimum read length threshold of 15 nucleotides [52]. Processed reads were aligned to the reference genome (GRCm38) using miRDeep2 [53]. The quantification of miRNAs was performed using miRDeep2 with miRBase v22 annotations [54]. Trimmed mean of M-values (TMM) normalization of miRNA read counts and differential expression analysis were performed using edgeR, version 3.36.0 [55]. TMM is a between-sample method that is suitable for comparing different libraries. The low variability of the TMM within a triplicate group confirms the quality of the RNA-seq data. For differential expression (DE) analysis, genes were filtered by requiring at least three samples to have a counts per million (CPM) value greater than four, with an FDR q-value < 0.05. The term ‘same’ marks changes in expression that are bounded by 33%. Partition of DEMs to clusters was performed according to the threshold listed in Appendix A.

The results of unsupervised clustering were performed using the R-base function “prcomp”. The analysis captures the maximum amount of variation in the data. Figures with added annotation were generated using the ggplot2 R package, version 3.3.5. All other statistical tests were performed using R-based functions. When appropriate, *p*-values < 0.05 were calculated and considered statistically significant. Results from microglial cell experiments are presented as the mean ± SD (standard deviation; see details in [15]).

The physical protein–protein interaction (PPI) set is based on STRING [56]. The STRING network was used based on a high PPI confidence score (>0.6). Connectivity networks excluded gene neighborhoods, gene fusion, and co-occurrence as evidence. miRNet 2.0 was used for analyzing miRNA functions through network-based visual analytics [57]. For functional analysis, the platform integrates miRNA with targets, transcription factors (TFs), and a knowledge-based graph [58]. The enrichment of miRNAs was based on miRinGO, which addresses indirect gene targets through transcription factors (TFs) according to miRNA expression in specific tissues [59]. The database dbDEMC 3.0 (database of differentially expressed miRNAs in human cancers) covers 40 cancer types with the large-scale compilation of miRNA gene expression from experiments [60].

## Figures and Tables

**Figure 1 ijms-26-05677-f001:**
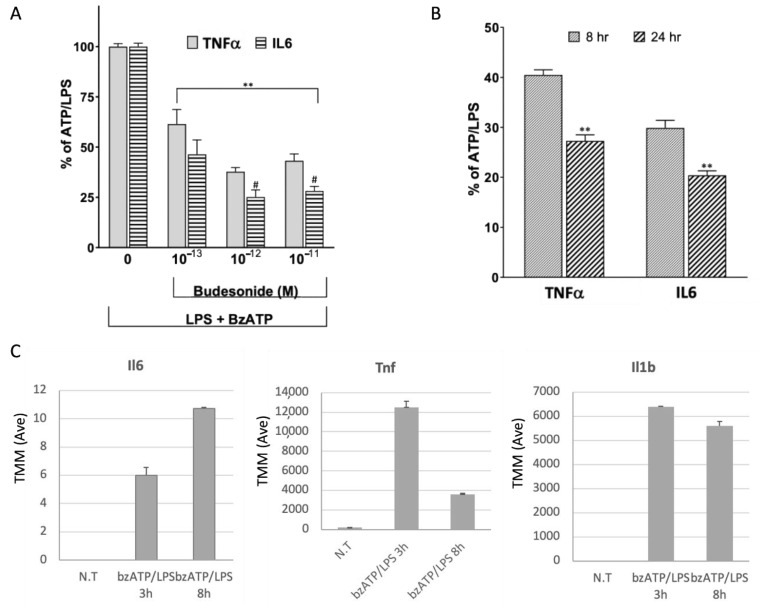
Quantitation of TNFα and IL-6 released from primary neonatal murine microglial culture following stimulation. The bars represent the related reduction in the release of TNFα and IL6 from microglia activated by BzATP/LPS. (**A**) A dose response relative to fully activated culture (100%). Budesonide was added 2 h before BzATP and LPS in the presence of 0.1% BSA. All concentrations of budesonide tested reduced the cytokines significantly (**, *p*-value < 0.01). Significant differences from the value for TNFα are indicated (# *p*-value < 0.05). (**B**) Difference in the reduction in the release of TNFα and IL6 after 8 h (as in (**A**), with a budesonide concentration of 10–12 M) and at 24 h. The difference between 8 and 24 h was statistically significant (**) with *p*-value < 0.01. (**C**) The level of transcripts for Il6, Tnf, and Il1b are tested from the results of RNA-seq analysis. The average TMM and the standard deviations from biological triplicates are shown for each experimental group. The statistical significance with the *p*-value FDR is very significant for Il6 (FDR 1.1 × 10^−18^), Tnf (FDR 2 × 10^−10^), and Il1b (FDR 1.3 × 10^−42^).

**Figure 2 ijms-26-05677-f002:**
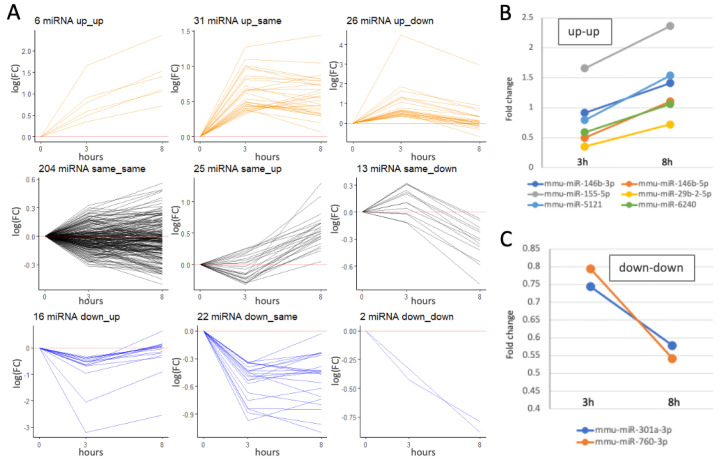
Trends in the gene expression of microglial miRNAs following the activation protocol with BzATP/LPS for 3 h and 8 h. The mapping of 372 miRNAs resulted in 345 uniquely labeled mature miRNAs, defined also by their 5p and 3p arms from the precursor pre-miRNAs. (**A**) Uniquely identified miRNAs (total 345) are classified into nine modules based on their combined expression trend (up, down, or same, and their combinations). For definitions and thresholds, see details in Appendix A. The baseline log2(FC) = 0 is shown by the horizontal red line. The miRNAs associated with a distinct short-term kinetics are colored for: up (orange), same (black), and down (blue). The number of miRNAs that belong to each module is indicated. (**B**) The fold change in the miRNAs that are labeled ‘up-up’. (**C**) The fold change in the miRNAs that are labeled ‘down-down’. The analysis is based on the data in Appendix A.

**Figure 3 ijms-26-05677-f003:**
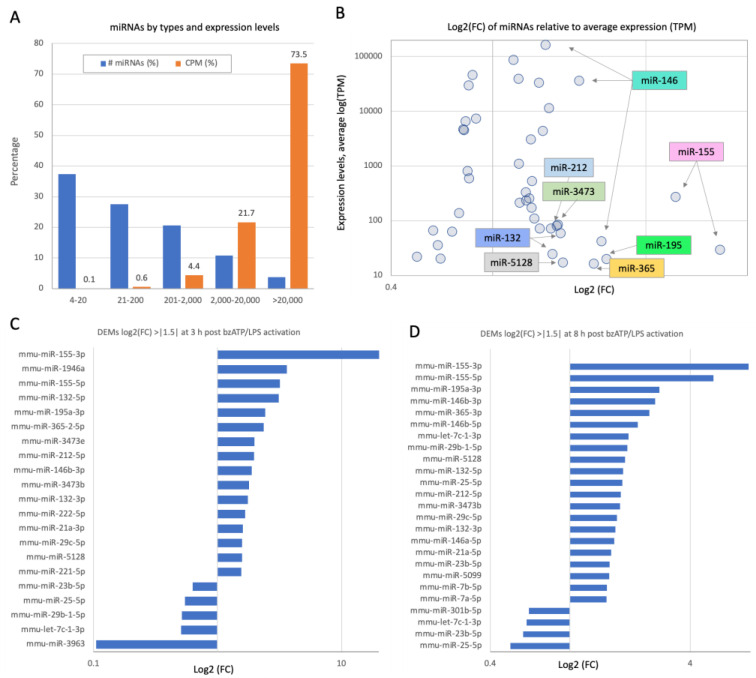
A global view of miRNAs’ statistics in microglia culture and time-dependent DEMs following activation. (**A**) Partition of the 372 identified miRNAs (in %) by their number of miRNAs (# miRNAs, blue) and by their expression level bins (orange, CPM). (**B**) Scatter plot of log2(FC) and the expression level (CPM). A few miRNAs with maximal FC and high expression levels are indicated. (**C**) A list of 21 significant DEMs 3 h post-activation with ≥|1.5| folds. (**D**) A list of 25 significant DEMs 8 h post-activation, with ≥|1.5| folds. The lists are sorted by the fold change. Only miRNAs that were differentially expressed by ≥1.5-fold (upregulated and/or downregulated). Note the larger breadth of the log2(FC) in the data collected from 3 h post-activation relative to 8 h post-activation. For a complete list, see Appendix A.

**Figure 4 ijms-26-05677-f004:**
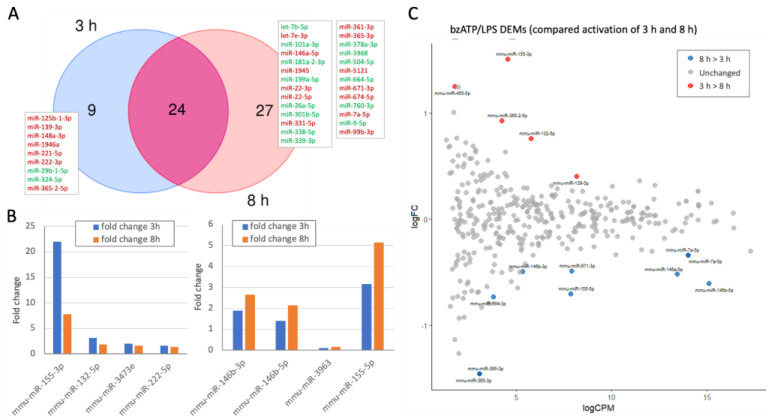
Dynamics of DEMs following the activation of microglial culture. (**A**) Venn diagram of the DEMs of two time points (3 h and 8 h). The unique sets that are not shared by both time points are listed. Included are all significant DEMs (FDR ≤ 0.05) and log2(FC) of >|0.33| relative to N.T. cells. The listed names of DEGs are colored as up- and downregulated DEMs by red and green font, respectively. (**B**) Based on the overlap in A, the representative listed DEMs are those displaying >30% difference in expression between 3 h and 8 h of cell activation. The listed DEMs show ‘up-down’ (**left**) and ‘up-up’ (**right**). (**C**) MA plot of the log2(FC) relative to the CPM (in log) for the 372 identified miRNAs. The temporal DEMs (T-DEMs) are colored by the change expression trends, with the expression of 8 h > 3 h in blue and that of 3 h > 8 h in red. Gray are miRNAs that showed no temporal expression pattern. The change in expression was determined by a threshold of log2(FC) > |0.33| and an FDR q-value < 0.05.

**Figure 5 ijms-26-05677-f005:**
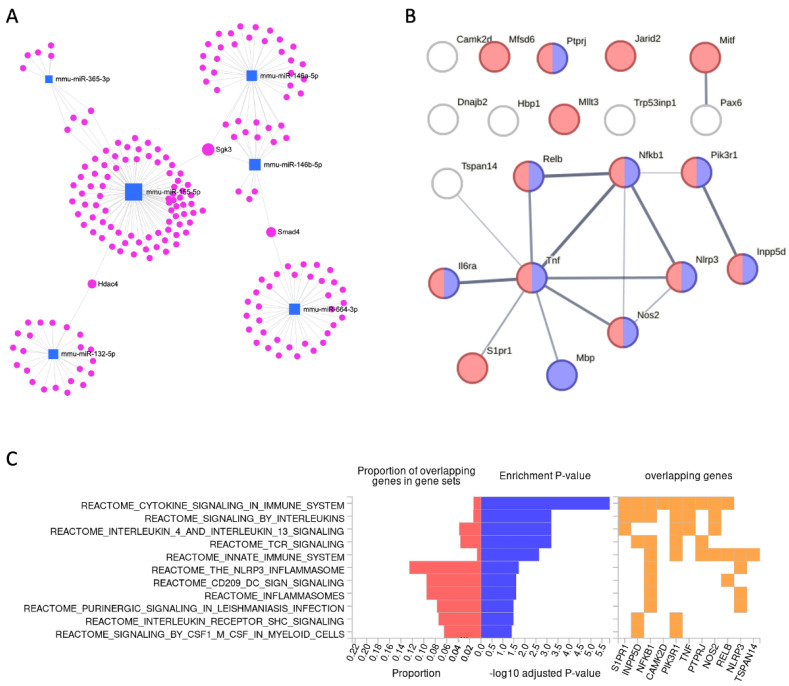
Network and functional view for T-DEMs and gene targets. (**A**) miRNet 2.0 view for the 6 identified T-DEMs (blue square) and their experimentally validated targets (pink marker, 244 genes). Genes connecting several miRNAs are labeled. The dominant connection of miR-155-5p is evident. (**B**) STRING view for a set of 21 protein-coding targets (T-DEGs, Table 1). The protein–protein interaction (PPI) network is significant (PPI enrichment *p*-value: 0.00986). The enrichment of the gene ontology biological process (GO_BP) of the immune system (GO:0002376) and the regulation of cytokine production (GO:0001817) are colored red and blue, respectively (*p*-value 3.4 × 10^−6^). Genes without these GO are not colored. (**C**) The functional enrichment of the listed genes (21 genes, Table 1) by Reactome pathways. From left to right: the fraction of gene input (red); significant pathways by Reactome are sorted by *p*-values (blue); and the overlapping genes for each of Reactome’s enriched pathways (orange).

**Figure 6 ijms-26-05677-f006:**
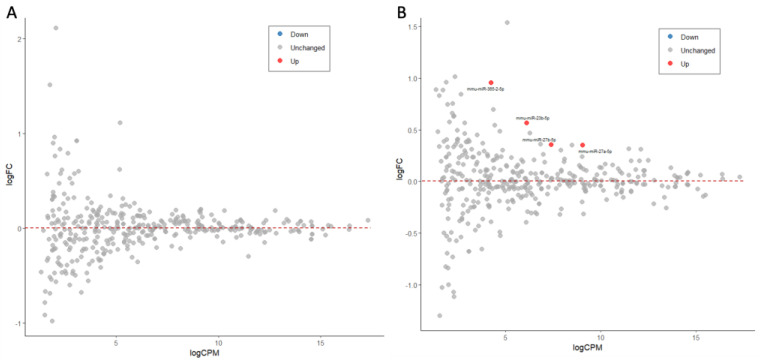
The effect of ladogtigil on miRNA expression. MA plots of the log2(FC) relative to the CPM (in log) for the 372 identified miRNAs in a fully activated setting (bzATP/LPS). (**A**) shows 3 h post-activation relative to N.T. Note that none of the miRNA was marked as significantly up (red) or down (blue). (**B**) shows 8 h post-activation relative to N.T. The miRNAs that were upregulated by ladostigil are marked in red font. The gray color is miRNAs that showed no significant expression change. The change in expression was determined by a threshold of log2(FC) > |0.33| and an FDR q-value < 0.05. The dashed lines mark stable expressions.

**Table 1 ijms-26-05677-t001:** T-DEMs and their experimental validated targets are based on bzATP/LPS microglia RNA-seq data.

Validated Target	Gene Name	miRNA ^a^ (E: 3 h > 8 h)	FDR T-DEG (3–8 h)	DEG Ave. CPM	FC T-DEG (3 h > 8 h)	FC DEG (8 h to N.T.)
*Camk2d*	Calcium/calmodulin-dependent protein kinase II, delta	**miR-146a-5p**	1.2 × 10^−18^	47.2	0.50	1.31
*Dnajb2*	DnaJ heat shock protein family (Hsp40) member B2	E: miR-155-3p	1.4 × 10^−15^	16.0	2.17	1.90
*Hbp1*	High mobility group box transcription factor 1	**miR-155-5p**	3.2 × 10^−14^	16.3	2.05	0.68
*Il6ra*	Interleukin 6 receptor, alpha	**miR-155-5p**	8.2 × 10^−17^	18.3	2.58	0.54
*Inpp5d*	Inositol polyphosphate-5-phosphatase D	**miR-155-5p**	5.2 × 10^−25^	178.1	2.20	1.04
*Jarid2*	Jumonji, AT rich interactive domain 2	**miR-155-5p**	6.5 × 10^−21^	35.0	0.42	2.29
*Mbp*	Myelin basic protein	**miR-7a-5p**	4.7 × 10^−21^	71.5	2.10	0.54
*Mfsd6*	Major facilitator superfamily domain containing 6	**miR-155-5p** miR-365-3p	4.9 × 10^−22^	116.0	2.06	1.35
*Mitf*	Melanogenesis associated transcription factor	E: miR-155-3p	1.3 × 10^−28^	354.5	0.34	1.05
*Mllt3*	Myeloid/lymphoid or mixed-lineage leukemia; translocated to, 3	**miR-146a-5p** miR-146b-5p	2.9 × 10^−14^	12.0	2.41	0.71
*Nfkb1*	Nuclear factor of kappa light polypeptide gene enhancer in B cells 1, p105	miR-146b-5p	3.6 × 10^−26^	523.3	0.43	2.59
*Nlrp3*	NLR family, pyrin domain containing 3	**miR-7a-5p**	1.6 × 10^−29^	1000.0	0.25	3.85
*Nos2*	Nitric oxide synthase 2, inducible	**miR-146a-5p**	6.0 × 10^−17^	22.6	3.10	39.13
*Pax6*	Paired box 6	**miR-7a-5p**	6.2 × 10^−14^	12.6	2.53	0.50
*Pik3r1*	Phosphoinositide-3-kinase regulatory subunit 1	E: miR-132-5p	1.2 × 10^−20^	47.4	2.16	0.81
*Ptprj*	Protein tyrosine phosphatase, receptor type, J	**miR-155-5p**	1.0 × 10^−22^	392.8	0.42	1.89
*Relb*	Avian reticuloendotheliosis viral (v-rel) oncogene related B	**miR-146a-5p**	3.5 × 10^−26^	56.2	0.34	1.10
*S1pr1*	Sphingosine-1-phosphate receptor 1	**miR-155-5p**	2.1 × 10^−15^	10.9	4.13	0.30
*Tnf*	Tumor necrosis factor	E: miR-132-5p	4.8 × 10^−28^	5250.7	0.29	18.92
*Trp53inp1*	Transformation related protein 53 inducible nuclear protein 1	**miR-155-5p**	1.1 × 10^−16^	18.3	3.24	0.57
*Tspan14*	Tetraspanin 14	**miR-155-5p**	9.2 × 10^−20^	41.9	2.32	0.94

^a^ Signified by temporal pattern. Early (E) indicates maximal expression at 3 h. The other miRNAs are signified as late (i.e., maximal expression at 8 h post-activation). In bold are the abundant miRNAs (average CPM > 200). Validated genes are alphabetically listed.

**Table 2 ijms-26-05677-t002:** Upregulation by ladostigil of miRNAs that act in reducing inflammation in multiple model systems.

miRNA	Activation/Condition	Cellular and Model ^a^	Target Gene/Pathway ^b^	Ref.
miR-23b-5p	Intracerebral hemorrhage (ICH)	Rat brain	IPMK	[28]
	Sepsis induced	Mouse	MMP/ADAM10	[29]
miR-27a-5p	LPS activated	Human dental pulp cells (hDPCs)	TAB1/NF-κB	[30]
	Clostridioides infection	Mouse intestine	NF-κB signaling	[31]
	Salmonella infection	EVs, RAW264.7 cells	TLR7/NF-κB	[32]
miR-27b-5p	TNF alpha exposure	Human aortic endothelial cells (HAECs)	Akt-FOXO1	[33]
	Mycobacterium infection	RAW264.7 cells	NF-κB signaling	[34]
	Hydrogen peroxide induced	RAW264.7 cells	NF-κB signaling	[35]
	IL-1β induced	Human osteoarthritis chondrocytes	MMP-13/NF-kB/p38	[36]

^a^ EVs, exosomal vesicles; RAW264.7 is a mouse-derived cell line of peripheral macrophages. SW-1353 is a human chondrosarcoma cell line. ^b^ IPMK, inositol polyphosphate multikinase; ADAM10, ADAM metallopeptidase domain 10; TAB1, TGF-beta activated kinase 1/MAP3K7 binding protein 1; and TLR7, Toll-like receptor 7.

## Data Availability

RNA-seq data files for miRNAs were deposited in ArrayExpress under the accession E-MTAB-14921.

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
