# Peer review of "Temporal Shifts in MicroRNAs Signify the Inflammatory State of Primary Murine Microglial Cells"

_ijms, 2025, doi:10.3390/ijms26125677_

Round 1
Reviewer 1 Report
Comments and Suggestions for Authors
ijms-3584768
Temporal shifts in microRNAs signify the inflammatory state of primary murine microglial cells
In this study, the authors examine changes in the miRNA transcriptomic profile upon stimulation with bzATP/LPS and in the presence of ladostigil as possible indicators of microglial inflammatory states.
However, the results are unclear. Therefore, the results presented in this paper require further organization and selection.
- Since the proinflammatory cytokines TNF-α and IL-6 were detected 24 h after activation, the temporal behavior of activated cells by the change in miRNA profiles should therefore include 24 h after exposure.
- To be consistent, the authors should also include in their study the analysis of the release of the proinflammatory cytokine IL-1β in response to microglia activation.
- The authors report that none of the 372 miRNAs identified were significantly altered by ladostigil. Therefore, the temporal behavior of cells activated by the change in miRNA profiles should include at least 8 and 24 hours after exposure.
- Table 3 summarizes the current knowledge on the link between the overexpression of miRNAs and their potential targets that drive inflammation reduction. Table 3 does not provide original results or explain the mechanism of miR-23b-5p, miR-27a-5p, miR-27b-5p and miR-365-2-5p activation in response to ladostigil.
- The Discussion section is incomplete and lacks depth. Furthermore, lines 372 to 380 should be moved from the Results section to the Discussion section. Furthermore, miR-3473e and miR-222 are not mentioned in the discussion.
- Lines 526-528 “Dysregulation of miRNAs, especially the most abundant ones, suggests they can serve as early biomarkers for oxidative stress-induced neuronal damage in NDDs, CNS disorders, and other brain pathologies”. The authors should explain this sentence: oxidative stress/inflammatory process.
- The authors should review the entire text, for example: TNF@; NF-@B...
- The section 3.2 (The alternation in miRNA profile in the presence of bzATP is minimal) should be included within the section 3.3 (Temporal expression miRNA profiles following bzATP/LPS activation).
- This reviewer cannot find the Supplementary Materials.
Author Response
Reviewer 1:
In this study, the authors examine changes in the miRNA transcriptomic profile upon stimulation with bzATP/LPS and in the presence of ladostigil as possible indicators of microglial inflammatory states. However, the results are unclear. Therefore, the results presented in this paper require further organization and selection.
Reply: Thank you for this valuable comment. We have revised the manuscript to present the results in a clearer and more structured manner, ensuring that the data are better organized and more focused to improve overall clarity and interpretation. We also provided a cleaner version (improved English/ removed typos).
Revised: We remove repetition (e.g., removed Table 1 to a supplemental Table, compressed original Fig. 3 for reducing repetition).
- Since the proinflammatory cytokines TNF-α and IL-6 were detected 24 h after activation, the temporal behavior of activated cells by the change in miRNA profiles should therefore include 24 h after exposure.
Reply: Thank you for your comment. We chose to analyze miRNA expression at earlier time points because miRNAs typically exhibit rapid transcriptional responses following stimulation (of course they are not going through ribosome translation or other post translation modification that are critical for proteins (as identifying cytokines after processing and secretion). The proinflammatory cytokines (TNF-α and IL-6) accumulate after secretion and therefore their measurement at 24 hours post-activation meets the cell response time line.
To clarify, we relaced the original Fig. 1 with a new one that better describe the sensitivity of the microglial culture. The new Fig. also show the expression level at the transcript levels for three major pro-inflammatory cytokines (Il6, Tnfa and Il1b).
- To be consistent, the authors should also include in their study the analysis of the release of the proinflammatory cytokine IL-1β in response to microglia activation.
Reply: Thank you for the suggestion. In our recent publications, we included detailed analysis of IL-1β, in addition to the other cytokines. We forward the reader to the details in our previous publication for primary microglial characterization. To clarify, in the revised version now relaced Fig. 1 and the associated text and showed also the expression level at the transcript levels for all three pro-inflammatory cytokines (Il6, Tnfa and Il1b).
- The authors report that none of the 372 miRNAs identified were significantly altered by ladostigil. Therefore, the temporal behavior of cells activated by the change in miRNA profiles should include at least 8 and 24 hours after exposure.
Reply: While no significant changes were observed at 3 hours after ladostigil, we note that at 8 hours after ladostigil exposure, only 4 miRNAs exhibited significant changes in expression. This suggests that ladostigil may have a delayed effect on miRNA regulation. This time dependency arguments were added to revised discussion.
- Table 3 summarizes the current knowledge on the link between the overexpression of miRNAs and their potential targets that drive inflammation reduction. Table 3 does not provide original results or explain the mechanism of miR-23b-5p, miR-27a-5p, miR-27b-5p and miR-365-2-5p activation in response to ladostigil.
Reply: Thank you for your comment. As Table 3 serves as a summary of existing knowledge. Note that we have added all relevant references to support miRNA inference with potential process. We claim that this small Table (called Table 2 in revised version) is useful to indicate the centrality of the Nf-kb relevance and other inflammation pathways.
- The Discussion section is incomplete and lacks depth. Furthermore, lines 372 to 380 should be moved from the Results section to the Discussion section. Furthermore, miR-3473e and miR-222 are not mentioned in the discussion.
Reply: Thank you for this constructive feedback. We have moved the discussion from Results to Discussion as requested. In revised Discussion we included specific commentary on miR-3473e and miR-222 to ensure a more comprehensive interpretation of our findings.
- Lines 526-528 “Dysregulation of miRNAs, especially the most abundant ones, suggests they can serve as early biomarkers for oxidative stress-induced neuronal damage in NDDs, CNS disorders, and other brain pathologies”. The authors should explain this sentence: oxidative stress/inflammatory process.
Reply: We revised the sentence. The claim we want to share suggests that the following miRNA expression is a lead to test biomarkers in brain pathologies and NDDs.
- The authors should review the entire text, for example: TNF@; NF-@B...
Reply: Thank you for pointing this out. We have carefully reviewed the entire manuscript and corrected typographical and formatting errors. Unfraternally, some of these mistakes resulted from reformatting of the manuscript (e.g., the symbol “@”).
- The section 3.2 (The alternation in miRNA profile in the presence of bzATP is minimal) should be included within the section 3.3 (Temporal expression miRNA profiles following bzATP/LPS activation).
Reply: Thank you for the suggestion. We have combined Section 3.2 with Section 3.3 as proposed to improve the flow and coherence of the results.
- This reviewer cannot find the Supplementary Materials.
Reply: Thank you for bringing this to our attention. It appears there was an issue with the initial upload of the Supplementary Materials. We have now re-uploaded the file to ensure it is accessible for review. We uploaded the supplemental Figures (S1-S2) and supplemental Tables (S1-S8) in a separated file.
Reviewer 2 Report
Comments and Suggestions for Authors
This study explores how miRNAs reflect the inflammatory activation states of microglia, the brain’s immune cells. Using primary microglial cultures from neonatal mice, the authors simulate neuroinflammatory conditions with bzATP and LPS to trigger activation, then track changes in miRNA expression at 3 and 8 hours. They find that specific miRNAs (e.g., miR-155, miR-132) are upregulated, while others like miR-3963 are downregulated in response to stimulation. They also test the anti-inflammatory drug ladostigil, finding that it modestly upregulates inflammation-suppressing miRNAs after 8 hours. The study suggests that miRNA profiles could serve as early biomarkers of neuroinflammatory states and aid in monitoring responses to therapeutics. Here are some of my comments:
- The manuscript becomes frequently dense with extended explanations that end up confusing the reader. I would suggest streamlining the results and focusing on only the most biologically meaningful miRNAs.
- Many sentences throughout the manuscript lack clarity regarding the experimental conditions being described. For example, in the results section, it is often unclear whether the authors are referring to bzATP alone, LPS, or a combination of both, and whether the data correspond to the 3-hour or 8-hour activation time point. This makes it difficult to interpret the findings accurately. The manuscript would benefit from a thorough re-reading to ensure that each sentence clearly specifies the treatment condition and time point being discussed.
- I think the study needs to justify the rationale behind using bzATP. Nowhere in the introduction section has the inflammatory stressors been mentioned, bzATP and LPS. And what is the reason for using both individually and together? Why do the authors think that minimal responses are observed in bzATP alone? And why do the authors end up normalizing the results to that condition, like in Figure 1?
- Figures in the manuscript lacks proper labelling which is making it difficult to understand the key takeaways, a simple subheading for the figures would be helpful.
- In the first section of results, the lines “The level of induction was >20 fold higher than that in the presence of bzATP. While the absolute level of IL6 was lower than that of TNF-α, the induction was approximately 50-100-fold for the combination of bzATP/LPS”. Is this talking about LPS treatment? Also, according to graphs, level of IL6 is around 60 and TNF is around 25. I don’t get why authors state that IL6 levels are lower.
- In the first section of results, authors describe monitoring mRNA transcript levels (IL6 transcript levels), but the data is not shown.
- The second results section mentions 372 identified miRNAs. Are these the entire list of miRNAs identified in primary microglia by sequencing?
- The entire paper is strong bioinformatics predictions but lacks experimental validations. This should be acknowledged as a limitation.
Author Response
Reviewer 2:
This study explores how miRNAs reflect the inflammatory activation states of microglia, the brain’s immune cells. Using primary microglial cultures from neonatal mice, the authors simulate neuroinflammatory conditions with bzATP and LPS to trigger activation, then track changes in miRNA expression at 3 and 8 hours. They find that specific miRNAs (e.g., miR-155, miR-132) are upregulated, while others like miR-3963 are downregulated in response to stimulation. They also test the anti-inflammatory drug ladostigil, finding that it modestly upregulates inflammation-suppressing miRNAs after 8 hours. The study suggests that miRNA profiles could serve as early biomarkers of neuroinflammatory states and aid in monitoring responses to therapeutics. Here are some of my comments:
- The manuscript becomes frequently dense with extended explanations that end up confusing the reader. I would suggest streamlining the results and focusing on only the most biologically meaningful miRNAs.
Reply: Thank you for your valuable suggestion. We have streamlined the Results section by concentrating on the most biologically relevant miRNAs and removing extended explanations to enhance clarity and readability (i.e., compressed original Fig. 3. Also, Table 1 was removed from main text and appears in a Supplemental Table).
- Many sentences throughout the manuscript lack clarity regarding the experimental conditions being described. For example, in the results section, it is often unclear whether the authors are referring to bzATP alone, LPS, or a combination of both, and whether the data correspond to the 3-hour or 8-hour activation time point. This makes it difficult to interpret the findings accurately. The manuscript would benefit from a thorough re-reading to ensure that each sentence clearly specifies the treatment condition and time point being discussed.
Reply: Thank you for this important observation. We have thoroughly revised the manuscript to clearly specify the treatment conditions (bzATP, LPS, or their combination) and the corresponding time points in each relevant section. This clarification improves the accuracy and ease of interpreting our findings.
- I think the study needs to justify the rationale behind using bzATP. Nowhere in the introduction section has the inflammatory stressors been mentioned, bzATP and LPS. And what is the reason for using both individually and together? Why do the authors think that minimal responses are observed in bzATP alone? And why do the authors end up normalizing the results to that
condition, like in Figure 1?
Reply: Thank. We forward the reader to previous publications that explained with great details the rationale for using bzATP and LPS as inflammatory stressors. Additionally, we clarify why bzATP alone elicits minimal responses. We forward the reader to previous papers to avoid repetition and refer to publications in which we established the working protocol.
Figures in the manuscript lacks proper labelling which is making it difficult to understand the key takeaways, a simple subheading for the figures would be helpful.
Reply: Thank you for your feedback. We have added clear titles and improved the labeling of all figures to enhance clarity and help readers easily grasp the key takeaways. We added titla to Figures for clarity.
- In the first section of results, the lines “The level of induction was >20 fold higher than that in the presence of bzATP. While the absolute level of IL6 was lower than that of TNF-α, the induction was approximately 50-100-fold for the combination of bzATP/LPS”. Is this talking about LPS treatment? Also, according to graphs, level of IL6 is around 60 and TNF is around 25. I don’t get why authors state that IL6 levels are lower.
Reply: Thank you for pointing this out. We have corrected the explanation to accurately reflect the data shown in the graphs. We removed the original Fig 1 and replaced with a new one to simplify the message and to provide missing information regarding the discussed cytokines. We also added the transcript level as asked (by another reviewer).
- In the first section of results, authors describe monitoring mRNA transcript levels (IL6 transcript levels), but the data is not shown.
This was tested and published in the past. We tried to stick to new finding only. Thus, we refer the readers to the data in which the transcripts of the IL^ are shown (using RT-PCR)
Reply: We relaced Fig. 1 to simplify the message and to provide missing information regarding the discussed cytokines. We also added the transcript level as asked (also by another reviewer).
- The second results section mentions 372 identified miRNAs. Are these the entire list of miRNAs identified in primary microglia by sequencing?
Reply: Thank you for your question. Yes, the 372 miRNAs mentioned represent the full set of miRNAs identified in primary microglia that passed the defined expression threshold in our sequencing analysis.
- The entire paper is strong bioinformatics predictions but lacks experimental validations. This should be acknowledged as a limitation.
Reply: Thank you for your comment. We agree with this important point and have now acknowledged that some of the literature-based support needs further validation.
Reviewer 3 Report
Comments and Suggestions for Authors
Microglia and neuroinflammation have been closely associated with neurodegenerative diseases. In the current study, Zohar et al. have presented a comprehensive microRNA (miRNA) profiling upon the bzATP and LPS treatment. A group of miRNAs was highlighted with significant changes and presented a temporal expression profiles closely related to bzATP/LPS activation. And interestingly, a neuroprotective compound can recover the differential expression of certain miRNA in a time-dependent manner. Overall, the study has provided resources to further examine the activation process and underscored the potential of miRNA as biomarkers. One minor edit is suggested as below:
- For Figure 1, the mRNA levels upon bzATP/LPS should also be included as supplementary data. Also, 8hr data is important to understand the activation process in the model and should be included as well.
Author Response
Reviewer 3:
Microglia and neuroinflammation have been closely associated with neurodegenerative diseases. In the current study, Zohar et al. have presented a comprehensive microRNA (miRNA) profiling upon the bzATP and LPS treatment. A group of miRNAs was highlighted with significant changes and presented a temporal expression profiles closely related to bzATP/LPS activation. And interestingly, a neuroprotective compound can recover the differential expression of certain miRNA in a time-dependent manner. Overall, the study has provided resources to further examine the activation process and underscored the potential of miRNA as biomarkers. One minor edit is suggested as below:
- For Figure 1, the mRNA levels upon bzATP/LPS should also be included as supplementary data. Also, 8hr data is important to understand the activation process in the model and should be included as well.
Reply: Thank you for the suggestion. We have included the 8-hour data to provide a more complete view of the activation process and have added the cytokine mRNA expression data following bzATP/LPS treatment as supplementary material as requested (revised new Fig. 1).
Round 2
Reviewer 1 Report
Comments and Suggestions for Authors
ijms-3584768
Temporal shifts in microRNAs signify the inflammatory state of primary murine microglial cells
Once the authors have made numerous improvements, the article could be considered for publication.

Author Response
Thanks for the positive response. We believe that the current revised version is simpler, smoother to read, and better organized. Thanks for the useful comments.
Reviewer 2 Report
Comments and Suggestions for Authors
Thank you to the authors for their valuable corrections and improving the manuscript. All of my previous queries are answered with satisfaction.
Author Response

(The authors gave the same response as above.)
